# The Impact of Family Life Cycle on Farmers' Living Clean Energy Adoption Behavior—Based on 1382 Farmer Survey Data in Jiangxi Province

Xiang Ding, Jing Wang and Shiping Li *

College of Economics and Management, Northwest A&F University, Xianyang 712100, China; dinglx96@nwafu.edu.cn (X.D.)
* Correspondence: lishiping68@nwafu.edu.cn

**Abstract:** Encouraging farmers to adopt greener and cleaner energy is crucial for reducing energy pollution and achieving carbon neutrality goals. In rural China, the decision making of farmers is often closely related to the whole family. At different stages of the family life cycle, the family has different characteristics, which leads to heterogeneity in the focus and final decision of farmers in adopting living clean energy. Therefore, this paper studies the farmers' living clean energy adoption behavior from the perspective of the family life cycle. It is helpful to identify the different policy needs and the evolution of farmers in different stages in order to provide a reference and inspiration for encouraging the adoption of living clean energy by farmers and for promoting the development of clean energy in rural areas. Based on the survey data of 1382 farmers in Jiangxi Province, this paper uses a multiple linear regression model to explore the impact of the family life cycle on farmers' clean energy adoption behavior. The results show the following: (1) The family life cycle has a significant impact on farmers' living clean energy adoption behavior, which is reflected in four aspects: energy demand, livelihood strategy, health demand and support burden; (2) Awareness of environmental ecology and frequency of government promotion have significant positive effects on farmers' living clean energy adoption behavior, while gender has significant negative effects on farmers' clean energy adoption behavior; (3) There are also differences in the influencing factors of farmers' living clean energy adoption behavior at different stages of the family life cycle. Therefore, when promoting clean energy in rural areas, a precise clean energy incentive mechanism should be adopted to treat families in different family life cycle stages differently.

**Keywords:** family life cycle; clean energy; farmers' adoption behavior; living; multiple linear regression





## 1. Introduction

At present, the problem of human survival, caused by climate and environmental change, has become a serious challenge facing the whole world. As the largest developing country and a major carbon dioxide emitter, China urgently needs to accelerate the development of a low-carbon and clean economy. On 22 September 2020, at the 75th session of the United Nations General Assembly, Xi Jinping, General Secretary of the Communist Party of China, announced that China would achieve carbon neutrality by 2060. Given that over 85% of carbon dioxide emissions come from energy activities, the carbon neutrality strategy is bound to profoundly change China's energy consumption structure [1]. China has more than 500 million rural farmers. With the development of China's rural economy, the increase in energy consumption and carbon emissions of rural residents is becoming the main growth point of China's energy consumption and carbon emissions [2]. Statistics show the 60% of the total rural energy consumption in China is attributed to rural living energy [3]. In the period between 1990 and 2019, the per capita living energy usage in rural areas increased from 83 kgce to 444 kgce [4]. The third national agricultural census

data reveal that the usage proportion of electricity and gas in China's rural living energy consumption witnessed an increase to 58.6% and 49.3%. While the proportion of traditional solid energy saw a slight decrease, with firewood accounting for 44.2% and coal accounting for 23.9%. It is not difficult to find that although the energy consumption structure of rural farmers has now improved significantly and shown a diversified trend, due to lifestyle habits, consumption costs and other reasons, some farmers in China still use solid energy as their main living energy source, and the proportion of solid energy, such as firewood and coal, is still not low [5]. The energy consumption structure of farmers in China is still unreasonable. The unreasonable structure of living energy use not only causes serious health risks [6] but also has many adverse effects on social and economic aspects, including employment and poverty reduction [7], economic welfare, education, gender equality [8], agricultural production, etc., thereby constraining the sustainable development of the social and economic development to a certain extent. Therefore, the cleaning of rural living energy will become a key link in China's energy structure adjustment, achieving carbon neutrality targets and improving the environment [9], and it is also of great significance to the sustainable development of society.

The 2021 and 2022 No.1 Central Document from China explicitly advocates for the robust development of clean energy in rural areas. As executors and beneficiaries of living clean energy adoption behavior, farmers' choices will directly affect the realization of carbon neutrality goals and environmental improvement. Therefore, it is even more necessary to study the farmers' living clean energy adoption behavior. Due to China being a "family-centered" society, farmers' behavior and decision making are often a reflection of the overall willingness of the family, which is greatly influenced by the family's resource endowment. And a family's resource endowment is closely related to the stage of the family life cycle. Therefore, conducting research on the influencing factors of farmers' living clean energy adoption behavior from the perspective of the family life cycle has a certain theoretical value and practical significance in order to delve into the characteristics and laws related to farmers' living clean energy adoption behavior and explore the driving and limiting factors of farmers' living clean energy adoption behavior.

## 2. Literature Review

Currently, numerous scholars have employed diverse methods to investigate the drives behind farmers' clean energy adoption behavior, which are primarily centered on personal characteristics, psychological aspects, psychological factors, and energy resource availability. (1) Concerning personal characteristics, income usually emerges as a pivotal factor. According to the Energy Ladder Hypothesis, as economic development levels and incomes improve, family life energy usage gradually transitions towards clean, efficient, and modern energy sources. Income is considered the most direct and crucial factor governing family energy choices according to this theory. A multitude of scholarly studies [10–13] has confirmed this position. The influence of non-farm employment on the farmers' energy adoption behavior is somewhat intricate, with the current impact's direction and extent remaining indistinct [14]. The majority of scholars [15–17] contend that an increase in the percentage of migrant workers in households leads to a more significant likelihood of farmers selecting clean energy options. Furthermore, personal characteristics, such as education level [18] and family size [19,20], have an impact on farmers' energy adoption behavior. (2) In terms of psychological factors, cognition [21], attitude [22], responsibility [23], and happiness [24] are all significant factors that impact farmers' energy adoption behavior. (3) Scholars [25,26] also suggest that geographical features, terrain features, and resource endowments also play a critical role in determining farmers' energy adoption behavior.

Existing research on the relationship between the family life cycle and farmer behavior mainly focuses on the following aspects: (1) Family life cycle and family labor force. With the evolution of the family life cycle, family population burden, employment choice [27] and labor-flow characteristics [28] have all changed; (2) Family life cycle and farmers' land-management behavior. Ficher et al. [29] and Liang et al. [30] analyzed the causal

relationship between the family life cycle and agricultural land transfer behavior based on the theory of the small-scale peasant economy. With the change of the family life cycle, land-use mode [31] and land management will also vary, and farmers will make land-transfer decisions according to the needs of different stages. Zhu et al. [32] found through empirical analysis that farmers' decision-making preferences for agricultural land transfer change from transfer into to transfer out as the family life cycle evolves; (3) Family life cycle and farmers' consumption behavior. The family life cycle is directly related to farmers' economic income [33], and the family has different social functions in different stages of the family life cycle, which makes the significant differences in the consumption structure, preference, and consumption pattern of farmers at different family life cycles [34].

The research thus far provides a theoretical foundation for this paper, but certain shortcomings must be addressed: (1) There is a lack of research into farmers' clean energy adoption behavior. Although there are some existing studies on farmers' energy adoption behavior, such as clean energy for cooking [35] and clean heating [36], these studies are not exactly the same in essence as farmers' living clean energy adoption behavior, nor are the variables completely consistent; (2) Although previous research indicates that household characteristics affect farmers' energy adoption behaviors, most studies only consider certain household characteristics as independent or control variables, disregarding the dynamic nature of household characteristics and the corresponding changes in farmers' behavior; (3) There is a lack of in-depth exploration of the mechanism of the family life cycle on farmers' living clean energy adoption behavior, and there is little literature on grouping farmers from the perspective of the family life cycle and exploring the differences in living clean energy adoption behavior among different groups. Hence, this paper uses the family life cycle, a composite variable divided based on information such as the number, quality, and population structure of family members to deeply discuss the impact mechanism of the family life cycle on farmers' living clean energy adoption behavior. Additionally, this paper investigates the factors that influence farmers' clean energy adoption behavior at different stages of the family life cycle. This paper can supplement existing research from the following aspects: (1) This paper focuses on the farmers' living clean energy adoption behavior in rural areas and uses the degree of adoption to comprehensively reflect their living clean energy adoption behavior. In addition, we divide farmers' living clean energy adoption behavior into farmers' cooking, heating, and bathing clean energy adoption behavior according to different energy use purposes and study the impact of the family life cycle on the three separately. This paper can expand the literature on farmers' living clean energy adoption behavior; (2) This paper improves the traditional family life cycle theoretical framework based on the actual situation in China. It not only considers the composite influences brought by different stages of the family life cycle but also specifically explains the specific influence mechanism of the family life cycle on farmers' living clean energy adoption behavior and verifies the differences in influencing factors among different types of families. This paper aims to serve as a reference for encouraging the adoption of living clean energy by farmers and for promoting the development of clean energy in rural areas.

## 3. Theoretical Analysis

### 3.1. Concept Definition

#### 3.1.1. Living Clean Energy

With the development of low-carbon concepts and the advancement of related technologies, the concept of living clean energy is still constantly changing and evolving with the development of low-carbon concepts and related technologies. Therefore, there is no agreed-upon definition of living clean energy, and scholars may have varying definitions of it depending on their research objects and focuses. Cleanliness is a relative concept that is measured in comparison to existing energy sources. So, determining whether an energy source is considered "clean" mainly depends on comparing it to the energy sources it replaces [37]. In this paper, based on relevant research and the actual situation in China's

rural areas, we define living clean energy as, compared to coal, wood fueled by straw and other energy sources that have occupied the main position in the past, the energy used by farmers with less pollution to the ecological environment in rural living. The living clean energy defined in this paper mainly includes electricity, solar energy, natural gas (including liquefied natural gas), and biogas.

3.1.2. Family Life Cycle and Its Division

The family life cycle is a sociological concept, which was first proposed by Rowntree in 1902 when he explained the causes of poverty [38]. It describes the development process of a family from birth, development, maturity, and decline to extinction [39]. Later, Glick [40], an American social demographer, proposed a relatively complete theory of the family life cycle for the first time in 1947. He divided the family life cycle into six stages, including formation, expansion, stability, contraction, empty nest, and disintegration. Since then, academics have made numerous revisions and advancements to this theory, including proposals for the 10 major stages and the 24 small-stage cycle models [41], the 9-stage model [42], the 8-stage model [43], the 13-stage model [44], etc. These revisions aim to achieve a more comprehensive understanding of the theory's principles and applications. It can be seen that although the family life cycle is an objective fact, most families will experience similar development trajectories, and scholars have significant differences in the classification standards, stages, and models of the family life cycle [45].

The main basis for the division of the family life cycle is whether the difference in the family at each stage is obvious [46]. First of all, due to variations in cultural and economic circumstances among different countries, the family life cycle exhibits different characteristics. For instance, in Chinese families heavily influenced by Confucian culture, several phenomena exist, such as "co-residence of multiple generations", "intergenerational child-rearing", and "intergenerational financial support" [47]. Secondly, the division of urban and rural area in China also leads to varied features of urban and rural families during different stages of the family life cycle. Furthermore, the composition of households in rural society has undergone changes due to societal progress and transformations. Therefore, this paper needs to divide the family life cycle that is suitable for the current rural environment and research objectives in China, ensuring that different family life cycles have obvious differences. Only in this way can the division and research have practical and theoretical significance.

The current classification of the family life cycle in China is mainly divided into two categories: one is based on the combination of family members and the age of their children, and the other is based on the demographic characteristics of the family [48]. This paper is inspired by Wang Wei et al.'s [49] approach to dividing the family life cycle of rural families. The family life cycle is divided into six stages based on population events, such as marriage, childbirth, adulthood, aging, and death, namely the initial stage, the raising stage, the burdening stage, the stable stage, the supported stage, and the empty-nest stage. The specific division is shown in Table 1.

**Table 1.** Rural family life cycle stage.

| Family Life Stage | Division Basis |
| --- | --- |
| Initial stage | Young couple without children |
| Raising stage | (Grand) Child born, youngest child under 18 years old, no elderly person over 65 years old |
| Burdening stage | (Grand) Child born, youngest child under 18 years old, elderly person over 65 years old |
| Stable stage | Children or grandchildren are all over 18 years old, and there are no elderly people aged 65 or above |
| Supported stage | Children or grandchildren are all 18 years old and have elderly people over 65 years old |
| Empty-nest stage | After separation, parents live alone |

### 3.2. Theoretical Analysis and Research Hypothesis

3.2.1. The Impact Mechanism of Family Life Cycle on Farmers' Living Clean Energy Adoption Behavior

The family life cycle is a reciprocating process from the formation to disintegration of a family, which is a comprehensive reflection of the characteristics of family human resources [32]. This paper divides the family life cycle based on family population characteristics. Therefore, with the evolution of the stages of the family life cycle, there are significant differences in the size, quality, and structure of the family population. Accordingly, the energy demand, livelihood strategy, health demand, and support burden of farmers in different family life cycles will also be different, thus, affecting the farmers' living clean energy adoption behavior (Figure 1).

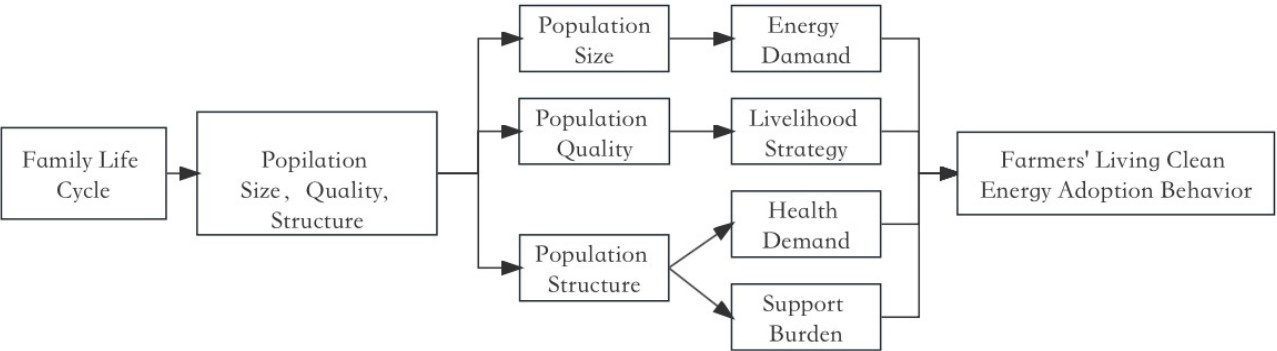

**Figure 1.** Theoretical framework for the influence of family life cycle on farmers' living clean energy adoption behavior.

Specifically, the evolution of the family life cycle leads to the increase or decrease of the family population, and the energy demand of farmers will also change accordingly, which will affect farmers' living clean energy adoption behavior. When farmers split their production and establish a new family, the family is smaller, and, therefore, their energy demand is also reduced. With the birth and growth of children and family mergers, the family size gradually increases, and the energy demand of farmers increases. However, traditional solid energy collection can no longer meet their daily needs. At this time, farmers will seek alternative energy sources, which may promote the adoption of clean energy in their daily lives.

There are differences in household population quality at different stages of the family life cycle, which can result in differing livelihood strategies for households and can influence farmers' adoption of clean energy in their daily lives. Firstly, from the perspective of opportunity cost, solid energy collection is a kind of household work without financial return [50]. Non-farm employment offers a higher return on investment compared to agricultural employment. If there are more non-farm employment opportunities, farmers will be more inclined to choose non-farml employment, consequently leading to less time spent collecting traditional solid energy. Farmers do not have enough time to collect traditional solid energy, making farmers seek more efficient and cleaner energy sources. Secondly, non-farm employment will cause farmers to reduce their land-management scale [51], affecting the planting area of crops, thereby reducing the traditional solid energy available to them. In order to meet their daily energy needs, non-farm employment may promote the adoption of clean energy by farmers [2]. Finally, non-farm employment will increase farmers' economic income, enabling them to pursue clean energy.

The development of the family life cycle is accompanied by changes in the family population structure, and the corresponding support burden and health demand will also be different, thereby affecting the farmers' living clean adoption behavior. First of all, variations in population support burden have an impact on household savings levels and subsequently affect farmers' investment decisions [52,53]. Compared with traditional solid energy, clean energy requires higher costs. When the support burden is heavy, more

household income is used for rigid expenses, which will inhibit farmers' clean energy adoption behavior in daily life. When the support burden is light, farmers can afford to use clean energy and will be more inclined to adopt clean energy in daily life. Secondly, with different family structures, the health status of rural households changes, leading to the different health needs of rural households. Farmers with different health needs will also have varying degrees of adoption of clean energy. For example, in the later stages of the family life cycle, as family members age, and the health status of families decreases to varying degrees, at this time the demand for health in the family will also increase, which will promote farmers to use clean energy that is more beneficial to the body compared to traditional solid energy.

In summary, different family life cycles are essentially the differences in the quantity, quality, and structure of the family population. In terms of clean energy adoption, differences in population quantity, quality, and structure can be deconstructed as differences in energy demand, livelihood strategy, support burden, and health demand, which can affect farmers' living clean energy adoption behavior.

3.2.2. Specific Characterizations of Different Family Life Cycle Stages and Analysis of Farmers' Living Clean Energy Adoption Behavior

(1) Initial stage. In this stage, family members are generally young couples who have recently separated from their original family and have not yet had any children. Families do not have children or elderly people to support, so the support burden is relatively light. However, non-farm employment has become the primary choice of occupation for most families. Hence, although the family size is small and the energy demand is low, their choice of non-farm employment makes them spend much less time collecting traditional solid fuels. In order to meet the daily energy needs of the family, farmers may choose cleaner modern energy.

(2) Raising stage. The demographic characteristics of the family in this stage are having children or grandchildren under the age of 18, and there are no elderly people aged 65 or above. With the birth of children and the growth in family size, the core task of families in this stage shifts to raising offspring. The responsibility for raising minors limits the full utilization of the female and paternal labor force. However, the average age of the core labor force is relatively low, and the level of human capital is relatively high. Non-farm employment remains the preferred option for most families. Due to the high cost of working outside and the many obstacles for children or grandchildren to study in different places, many families at this stage tend to implement vertical intergenerational division of labor or horizontal marital division of labor [54]. As a result, the whole family shows the characteristics of part-time employment. Compared with families in the initial stage, raising-stage families have accumulated a certain amount of economic capital and have the ability to support their adoption of clean energy. In addition, an increase in the size of families leads to an overall increase in energy demand. In order to provide a more comfortable and healthier growth environment for minors, farmers will be more inclined to use living clean energy.

(3) Burdening stage. The demographic characteristics of families in this stage are that the youngest child or grandchild is under 18 years old and there is an elderly person aged 65. With the birth of children and aging parents, families bear the responsibility of raising children and supporting the elderly and are in an "elderly at the top, young at the bottom" stage. Most families usually opt for part-time or non-farm employment as their primary livelihood strategy. At this stage, the support burden and economic pressure gradually increase, and farmers will be more inclined to control the cost of living, which will suppress the farmers' living clean energy adoption behavior.

(4) Stable stage. The demographic characteristics of families in this stage are that the youngest child or grandchild is over 18 years old, and there are no elderly people over 65 years old. At this stage, the economic burden of farmers is relatively light, and the level of human capital is relatively high. Young and middle-aged people go out

to work more, while their parents stay at home to continue agricultural production. Therefore, the people who stay in rural areas are older, their health level begins to decline, and the demand for health increases. At the same time, farmers have sufficient capital to use clean energy, so they are more inclined to use living clean energy.

(5) Supported stage. The demographic characteristics of families in this stage are that all children and grandchildren are over 18 years old, and there are elderly people over 65 years old. Compared to the stable stage, the support burden of the family increases during this stage, the level of human capital decreases, and the livelihood strategy of families gradually leans towards being agriculture-oriented [55]. At this stage, as family members age, the demand for health also increases, which will encourage farmers to adopt living clean energy. But at the same time, the support burden of the family increases during this stage, and the economic level will decrease because the livelihood strategy will gradually focus on agriculture. This in turn will inhibit farmers from using clean energy, so there is uncertainty about whether such farmers will adopt living clean energy.

(6) Empty-nest stage. The demographic characteristics of families in this stage are that all family members are over 65 years old. At this stage, households possess a comparatively low level of human capital. Although there is a high demand for health, the economic level is relatively low. Therefore, the farmers' adoption of living clean energy for daily use will be impeded by economic capital. And at this stage, most family members have experienced energy scarcity, so they will be more economical in energy use and reduce energy demand [56]. Overall, during empty-nest stage, the probability of farmers implementing living clean energy adoption behavior is relatively low.

In summary, there are differences in farmers' living clean energy adoption behaviors among different family life cycle stages. Among them, the probability of adopting living clean energy is higher for farmers in their initial stage, raising stage, and stable stage, but it is lower for farmers in their burdening stage and empty-nest stage. For farmers in their supported stage, there is uncertainly regarding the probability of adopting living clean energy.

## 4. Model Construction, Variable Selection, and Data Sources

*4.1. Model Construction*

In this paper, the farmers' living clean energy adoption behavior is a continuous variable. Therefore, the multiple linear regression model is applied to estimate. The general form of the multiple linear regression model is

$$Y_i^* = \alpha_0 + \alpha_1 X_i + \alpha_2 C_i + \varepsilon_i \tag{1}$$

In Formula (1), $Y_i^*$ is the degree of farmers' living clean energy adoption behavior of the i-th farmer, $X_i$ is the stage of family life cycle of the i-th farmer, $C_i$ is the control variable that affects the farmers' living clean energy adoption behavior of the i-th farmer, $\alpha_0$, $\alpha_1$, $\alpha_2$ is the parameter to be estimated, and $\varepsilon_i$ is the random disturbance term of the i-th farmer.

*4.2. Variable Selection*

4.2.1. Dependent Variable

In this paper, the dependent variable is the farmers' living clean energy adoption behavior. According to the above definition, living clean energy refers to an energy source that produces less pollution than the energy used in the past. However, in some areas, energy use may not be cleaner than the previous lifestyle (such as electricity used for daily lighting, television, etc. or energy used for motor vehicles, generators, etc.). Therefore, in order to address this issue, this paper mainly measures the farmers' living clean energy adoption behavior from three aspects: cooking, heating, and bathing [5]. After referring to the relevant literature [57], the farmers' living clean energy adoption behavior is defined as

the process of farmers using clean energy (electricity, solar energy, natural gas (including liquefied natural gas) and biogas) to meet their own cooking, heating, and bathing needs. In the empirical process, it is characterized by the degree of living clean energy adoption by farmers. The specific examination is conducted by asking the following questions: "How often do you use electricity in the cooking process?", "How often do you use solar energy in the cooking process?", "How often do you use natural gas in the cooking process?", "How often do you use biogas in the cooking process?", "How often do you use electricity in the heating process?", "How often do you use solar energy in the heating process?", "How often do you use natural gas in the heating process?", "How often do you use biogas in the heating process?", "How often do you use electricity in the bathing process?", "How often do you use solar energy in the bathing process?", "How often do you use natural gas in the bathing process?", "How often do you use biogas in the bathing process?" The degree of living clean energy adoption is represented by the values obtained after the total average of the scores for these 12 questions. These 12 questions are all on a Likert Level-5 scale.

### 4.2.2. Core Independent Variable

The core independent variable of this paper is the family life cycle. In existing research on the impact of the family life cycle, most scholars directly use each stage of the family life cycle as an independent variable, such as Zhang [58], Ye [48], etc. But this approach cannot effectively reflect the specific impact of the family life cycle on farmers' living clean energy adoption behavior. Therefore, according to the classification in Table 1, this paper firstly codes the stages of different farmers' family life cycle based on the number, age, structure, and marital status of family members and constructs dummy variables of the raising stage, burdening stage, stable stage, supported stage and 4 families with empty-nest stage as the benchmark.

Subsequently, based on the theoretical analysis above, this paper includes household characteristic variables to reflect the specific performance of different family life cycles in terms of farmers' living adoption behavior. These variables include energy demand, livelihood strategy, health demand, and support burden. Firstly, energy demand is reflected by asking the surveyed farmers the following question: "What do you think of your annual energy consumption level?" The greater their energy consumption, the higher their energy demand, and, thus, the more likely they are to adopt living clean energy. Secondly, non-farm employment reflects the livelihood strategy, which is primarily measured by the proportion of non-farm workers to the total number of households. The higher the degree of non-farm employment, the more likely farmers are to adopt living clean energy. Thirdly, health demand is reflected by farmers' self-assessment of their health status. This was achieved by asking those surveyed the following question: "How would you rate your current health status compared to your peers?" The lower the farmers' self-health evaluation, the higher their health demand, and, thus, the greater their probability of adopting living clean energy. Fourthly, support burden is reflected by the proportion of total annual household expenses to total income. The higher the proportion of total household expenses to total income, the lower the probability of farmers adopting living clean energy.

### 4.2.3. Control Variables

Drawing on existing research findings, this paper categorizes the control variables that may affect farmers' living clean energy adoption behavior into three categories. The first one is personal characteristics, including the gender, age, and education level of the surveyed farmers. The second is cognitive factors, mainly including the surveyed farmers' awareness of environmental ecology and their awareness of living clean energy. The third is policy factors, mainly including the frequency of government promotion of clean energy and the satisfaction of farmers with existing clean energy promotion policies. The specific variable definition, assignment, and descriptive statistical analysis are shown in Table 2.

**Table 2.** Variable setting and assignment description.

| Variables | Variable Definition and Assignment | Mean Value | Standard Deviation | Expected Direction |
|---|---|---|---|---|
| Dependent variable | | | | |
| Farmers' living clean energy adoption behavior (B) | Farmers' clean energy adoption behaviors—comprehensive value | 2.678 | 1.055 | |
| Core independent variable | | | | |
| Family life cycle | | | | |
| Raising stage (RS) | In the raising stage = 1; Others = 0 | 0.320 | 0.468 | + |
| Burdening stage (BS) | In the burdening stage = 1; Others = 0 | 0.380 | 0.487 | − |
| Stable stage (STS) | In the stable stage = 1; Others = 0 | 0.150 | 0.353 | + |
| Supported stage (SUS) | In the supported stage = 1; Others = 0 | 0.110 | 0.314 | ? |
| Empty-nest stage (reference group) (ES) | 0 | 0.030 | 0.166 | |
| Energy demand | | | | |
| Energy consumption in last year (ED) | Very low = 1; Relatively low = 2; Generally = 3; Relatively high = 4; Very high = 5 | 3.630 | 0.979 | + |
| Livelihood strategy | | | | |
| Non-farm employment (LS) | The proportion of non-farm workers to the total number of households. | 0.366 | 0.259 | + |
| Health demand | | | | |
| Health status (HD) | Very good = 1; Good = 2; Generally = 3; Bad = 4; Very bad = 5 | 2.339 | 0.907 | + |
| Support burden | | | | |
| Ratio of expenditure to income (SB) | The proportion of total annual household expenses to total income | 0.538 | 0.459 | − |
| Control variables | | | | |
| Personal characteristics | | | | |
| Gender (X1) | Male = 1; Female = 0 | 0.700 | 0.458 | − |
| Age (X2) | Unit: year | 45.32 | 12.199 | ? |
| Education level (X3) | Never go to school = 1; Primary school = 2; Middle school = 3; High school = 4; College or above = 5 | 3.480 | 1.000 | + |
| Cognitive factors | | | | |
| Awareness of environmental ecology (X4) | Very ignorant = 1; Basic ignorance = 2; Generally = 3; Basic understanding = 4; Well understood = 5 | 3.560 | 1.126 | + |
| Awareness of clean energy (X5) | Very ignorant = 1; Basic ignorance = 2; Generally = 3; Basic understanding = 4; Well understood = 5 | 3.940 | 0.877 | + |
| Policy factors | | | | |
| Frequency of government promotion of clean energy (X6) | Never = 1; Less frequently = 2; Generally = 3; Relatively frequent = 5; Always = 5 | 3.180 | 0.941 | + |
| Satisfaction of farmers with existing clean energy promotion policies (X7) | Very dissatisfied = 1; Dissatisfied = 2; Generally = 3; Satisfied = 4; Very satisfied = 5 | 4.020 | 0.983 | + |

### 4.3. Data Sources and Basic Characteristics of Samples

The data come from a field survey conducted in 2022 in rural areas of Jiangxi Province. This survey used the method of stratified step-by-step sampling and random sampling to select sample farmers. The specific procedure was as follows: based on the economic development situation of the region, all counties (cities) in Jiangxi Province were divided into three levels. Nanchang, Ganzhou, Jiujiang, and Shangrao are the first level; Yichun, Ji'an, Fuzhou, and Pingxiang are the second level; and Xinyu, Jingdezhen, and Yingtan are the third level. Then, five sample counties were randomly selected from each level, and one to five sample villages were randomly selected from each county. Farmers in each sample village were randomly selected for a "one-to-one" interview questionnaire survey. The content of the questionnaire included individual characteristics of farmers, family characteristics, production and living characteristics, and energy consumption behavior. A total of 1500 questionnaires were sent out, and 1460 were collected, with a collection

rate of 97.33%. After excluding those with information loss and logical inconsistencies, 1382 questionnaires were finally valid, with an effective rate of 94.66%. The basic characteristics of the samples are shown in Table 3.

It can be seen that 70.043% of the surveyed samples are males. This is mainly because the main object of this survey is the head of each family, resulting in a higher proportion of males. The education level of the farmers in the surveyed sample is mainly junior high school or below, accounting for 55.499%. The overall kurtosis of education level is $-0.541$, and the skewness is $-0.034$. The age of the farmers is mainly between 36 and 50 years old, accounting for almost half of the sample. The overall kurtosis of age is $-0.500$, and the skewness is 0.014. The family size is mainly composed of 3–4 people and 5–6 people, accounting for 45.166% and 43.343% of the sample, respectively. The overall kurtosis of family size is 2.622, and the skewness is 0.828. The number of farmers with only agricultural activities is 746, more than half of the sample. In summary, the basic characteristics of the sample farmers are basically consistent with the current rural situation in China, so the sample farmers have a certain degree of representativeness.

According to the family life cycle stages divided in the previous text, the results of classification and statistical analysis of the samples are shown in Table 4. From the perspective of sample composition, the number of households in the burdening stage is the highest, with a total of 532 households representing 38.495% of the total number of households surveyed, followed by 448 households in the raising stage, representing 32.417%. The lowest number is 8 households in the initial stage, which is only 0.579%. This may be due to the fact that the phenomenon of "splitting households without splitting them" is very common in current rural areas. Young couples are less likely to live alone with their parents, and young couples are more likely to go out to work. Therefore, in order to ensure the representativeness of the sample, in the empirical process of this paper we temporarily do not consider families in the initial stage and select only families in four stages—the raising stage, the burdening stage, the stable stage, and the supported stage—as the empirical objects. From the perspective of the average level of farmers' living clean energy adoption behavior in different life cycle stages, the highest level of farmers' living clean energy adoption behavior is the stable stage, with 3.324. Secondly, the raising stage and the supported stage are relatively close, with 3.046 and 3.003, respectively. The lowest average level of farmers' living clean energy adoption behavior is the burdening stage, with 2.045.

**Table 3.** Basic characteristics of samples.

| Characteristics | Description | Frequency | Percentage (%) | Kurtosis | Skewness | Characteristics | Description | Frequency | Percentage (%) | Kurtosis | Skewness |
|---|---|---|---|---|---|---|---|---|---|---|---|
| Gender | Male | 968 | 70.043 | −1.234 | −0.876 | | Age 35 and under | 311 | 22.504 | | |
| | Female | 414 | 29.957 | | | | 36–50 years old | 603 | 43.632 | | |
| | Never go to school | 30 | 2.171 | | | Age | 51–65 years old | 405 | 29.305 | −0.500 | 0.014 |
| | Primary school | 157 | 11.360 | | | | Age 66 and older | 63 | 4.559 | | |
| Education level | Middle school | 580 | 41.968 | −0.541 | −0.034 | | 1–2 people | 56 | 4.052 | | |
| | High school | 344 | 24.891 | | | | 3–4 people | 486 | 45.166 | | |
| | College or above | 271 | 19.609 | | | Family size | 5–6 people | 599 | 43.343 | 2.622 | 0.828 |
| Only engaged in agricultural work | Yes | 746 | 53.980 | −1.980 | −1.151 | | 7 people and above | 241 | 17.438 | | |
| | No | 636 | 46.020 | | | | | | | | |

**Table 4.** Composition of samples in various family life cycle stages.

| Family Characteristics | Initial Stage | Raising Stage | Burdening Stage | Stable Stage | Supported Stage | Empty-Nest Stage |
|---|---|---|---|---|---|---|
| Frequency | 8 | 448 | 532 | 202 | 153 | 39 |
| Percentage (%) | 0.579 | 32.417 | 38.495 | 14.616 | 11.071 | 2.822 |
| Energy demand | 3.750 | 3.650 | 3.538 | 3.639 | 3.595 | 3.256 |
| Non-farm employment | 0.500 | 0.332 | 0.326 | 0.464 | 0.393 | 0.675 |
| Ratio of expenditure to income | 0.517 | 0.592 | 0.550 | 0.515 | 0.474 | 0.579 |
| Health demand | 1.875 | 2.292 | 2.248 | 2.381 | 2.582 | 2.923 |
| Gender | 0.875 | 0.665 | 0.720 | 0.678 | 0.745 | 0.744 |
| Age | 36.625 | 43.069 | 45.545 | 44.525 | 49.366 | 58.026 |
| Education level | 2.750 | 3.571 | 3.494 | 3.574 | 3.386 | 2.410 |
| Awareness of environmental ecology | 3.625 | 3.605 | 3.511 | 3.515 | 3.667 | 3.462 |
| Awareness of clean energy | 4.000 | 3.920 | 4.006 | 3.787 | 4.033 | 3.769 |
| Frequency of government promotion of clean energy | 3.125 | 3.192 | 3.102 | 3.168 | 3.268 | 2.769 |
| Satisfaction of farmers with existing clean energy promotion policies | 4.125 | 3.998 | 4.118 | 3.946 | 3.941 | 3.538 |
| Farmers' living clean energy adoption behavior | 2.541 | 3.046 | 2.045 | 3.324 | 3.003 | 2.487 |

## 5. Empirical Analysis

In order to verify whether the household life cycle has a significant impact on farmers' living clean energy adoption behavior, this paper uses SPSS 26.0 software to perform multiple linear regression on the data. Before the regression analysis, in order to increase the reliability of the conclusion, the robustness of the model is tested. First is the multicollinearity diagnosis, which is tested by calculating the correlation coefficient, the variance inflation factor (VIF), and the tolerance of each variable in the model. In general, if the correlation coefficient is less than 0.8 [59], the tolerance is greater than 0.1, and if VIF is less than 10, it is considered reasonable, and there is no collinearity [60]. After testing, the correlation coefficient is less than 0.8 (Table 5). The VIF of each variable in this model is less than 10. Except for the raising stage and the burdening stage, the VIF values of the other variables are all less than 5, and the tolerance is greater than 0.1. This indicates that the model passed the multicollinearity test. The second approach is to gradually add control variables. Model 1 added only core independent variables of the family life cycle, while Model 2, Model 3, and Model 4 gradually added personal characteristics, cognitive factors, and policy factors on the basis of Model 1. Table 6 shows that with the gradual addition of control variables, the adjusted $R^2$ value of the model also gradually increases, and most of the key variables in the four models are significant and consistent in direction, indicating that the research conclusion has strong robustness. In addition, the *p*-values of all four models were 0.000, indicating that the model passed the 1% significance test. The $R^2$ values were 0.279, 0.288, 0.307, and 0.352, respectively. This indicates that the overall goodness of fit of the model meets the requirements and is statistically significant.

**Table 5.** Correlation coefficient of independent variables.

| | RS | BS | STS | SUS | ED | LS | HD | SB | X1 | X2 | X3 | X4 | X5 | X6 | X7 |
|---|---|---|---|---|---|---|---|---|---|---|---|---|---|---|---|
| RS | 1 | −0.548 ** | −0.287 ** | −0.244 ** | 0.015 | −0.092 ** | −0.036 | 0.081 ** | −0.053 * | −0.128 ** | 0.061 * | 0.025 | −0.018 | 0.008 | −0.014 |
| BS | −0.548 ** | 1 | −0.327 ** | −0.279 ** | 0.010 | −0.124 ** | −0.071 ** | −0.035 | 0.034 | 0.015 | 0.008 | −0.028 | 0.057 * | 0.001 | 0.081 ** |
| STS | −0.287 ** | −0.327 ** | 1 | −0.146 ** | 0.004 | 0.156 ** | 0.019 | −0.022 | −0.020 | −0.027 | 0.037 | −0.018 | −0.073 ** | −0.006 | −0.031 |
| SUS | −0.244 ** | −0.279 ** | −0.146 ** | 1 | −0.012 | 0.036 | 0.094 ** | −0.050 | 0.034 | 0.117 ** | −0.035 | 0.032 | 0.036 | 0.033 | −0.028 |
| ED | 0.015 | 0.010 | 0.004 | −0.012 | 1 | −0.140 ** | −0.066 * | −0.030 | −0.018 | −0.106 ** | 0.134 ** | 0.095 ** | 0.364 ** | 0.236 ** | 0.215 ** |
| LS | −0.092 ** | −0.124 ** | 0.156 ** | 0.036 | −0.140 ** | 1 | 0.043 | −0.082 ** | 0.092 ** | 0.161 ** | −0.166 ** | −0.055 * | −0.086 ** | −0.088 ** | −0.083 ** |
| HD | −0.036 | −0.071 ** | 0.019 | 0.094 ** | −0.066 * | 0.043 | 1 | −0.032 | −0.036 | 0.314 ** | −0.358 ** | 0.051 | −0.058 * | −0.018 | −0.065 * |
| SB | 0.081 ** | −0.035 | −0.022 | −0.050 | −0.030 | −0.082 ** | −0.032 | 1 | −0.006 | −0.030 | 0.011 | −0.023 | −0.073 ** | −0.088 ** | −0.060 * |
| X1 | −0.053 * | 0.034 | −0.020 | 0.034 | −0.018 | 0.092 ** | −0.036 | −0.006 | 1 | 0.023 ** | 0.007 | 0.001 | 0.033 | −0.056 * | −0.004 |
| X2 | −0.128 ** | 0.015 | −0.027 | 0.117 ** | −0.106 ** | 0.161 ** | 0.314 ** | −0.030 | 0.023 ** | 1 | −0.458 ** | 0.009 | −0.080 ** | −0.092 ** | −0.052 |
| X3 | 0.061 * | 0.008 | 0.037 | −0.035 | 0.134 ** | −0.166 ** | −0.358 ** | 0.011 | 0.007 | −0.458 ** | 1 | −0.065 * | 0.113 ** | 0.175 ** | 0.109 ** |
| X4 | 0.025 | −0.028 | −0.018 | 0.032 | 0.095 ** | −0.055 * | 0.051 | −0.023 | 0.001 | 0.009 | −0.065 * | 1 | 0.139 ** | 0.177 ** | 0.313 ** |
| X5 | −0.018 | 0.057 * | −0.073 ** | 0.036 | 0.364 ** | −0.086 ** | −0.058 * | −0.073 ** | 0.033 | −0.080 ** | 0.113 ** | 0.139 ** | 1 | 0.371 ** | 0.321 ** |
| X6 | 0.008 | 0.001 | −0.006 | 0.033 | 0.236 ** | −0.088 ** | −0.018 | −0.088 ** | −0.056 * | −0.092 ** | 0.175 ** | 0.177 ** | 0.371 ** | 1 | 0.339 ** |
| X7 | −0.014 | 0.081 ** | −0.031 | −0.028 | 0.215 ** | −0.083 ** | −0.065 * | −0.060 * | −0.004 | −0.052 | 0.109 ** | 0.313 ** | 0.321 ** | 0.339 ** | 1 |

Note: * and ** are statistically significant at the level of 1% and 5% respectively.

**Table 6.** Model results for the influence of family life cycle on farmers' living clean energy adoption behavior.

| Variable | Model 1 | | Model 2 | | Model 3 | | Model 4 | |
|---|---|---|---|---|---|---|---|---|
| | B | S.E. | B | S.E. | B | S.E. | B | S.E. |
| RS | 0.660 *** | 0.141 | 0.602 *** | 0.142 | 0.599 *** | 0.14 | 0.573 *** | 0.136 |
| BS | −0.035 * | 0.14 | −0.395 ** | 0.141 | −0.398 ** | 0.139 | −0.413 ** | 0.135 |
| STS | 0.871 *** | 0.147 | 0.802 *** | 0.148 | 0.819 *** | 0.146 | 0.787 *** | 0.141 |
| SUS | 0.549 *** | 0.152 | 0.500 *** | 0.152 | 0.477 *** | 0.15 | 0.434 *** | 0.145 |
| ED | 0.140 *** | 0.025 | 0.134 *** | 0.025 | 0.089 *** | 0.026 | 0.070 ** | 0.026 |
| LS | 0.322 ** | 0.098 | 0.382 *** | 0.099 | 0.413 *** | 0.098 | 0.426 *** | 0.095 |
| HD | 0.105 *** | 0.027 | 0.123 *** | 0.029 | 0.120 *** | 0.029 | 0.105 *** | 0.028 |
| SB | −0.223 *** | 0.053 | −0.221 *** | 0.053 | −0.202 *** | 0.052 | −0.172 *** | 0.05 |
| X1 | | | −0.203 *** | 0.055 | −0.215 *** | 0.054 | −0.182 *** | 0.052 |
| X2 | | | 0.002 | 0.002 | 0.003 | 0.002 | 0.002 | 0.002 |
| X3 | | | 0.081 ** | 0.029 | 0.087 ** | 0.028 | 0.052 | 0.028 |
| X4 | | | | | 0.101 *** | 0.021 | 0.083 *** | 0.022 |
| X5 | | | | | 0.107 *** | 0.029 | 0.032 | 0.03 |
| X6 | | | | | | | 0.270 *** | 0.028 |
| X7 | | | | | | | −0.051 | 0.027 |
| Constant term | 1.657 *** | 0.193 | 1.430 *** | 0.251 | 0.769 ** | 0.27 | 0.681 ** | 0.263 |
| Adjusted R2 | 0.279 | | 0.288 | | 0.307 | | 0.352 | |
| P | 0 | | 0 | | 0 | | 0 | |

Note: B is the regression coefficient of the variable; *, **, and *** are statistically significant at the level of 1%, 5%, and 10%, respectively.

*5.1. The Impact of Family Life Cycle on Farmers' Living Clean Energy Adoption Behavior*

From the perspective of the composite variables of the family life cycle, it can be seen that each family life cycle has a significant impact on farmers' living clean energy adoption behavior, which is consistent with the theoretical prediction direction and descriptive statistical analysis direction mentioned above (Table 6). The family in the raising stage has a significant positive impact on farmers' living clean energy adoption behavior, with disease being significant at a 1% level. It indicates that farmers during the raising stage are willing to adopt living clean energy, and their adoption level is significantly higher than that of the empty-nest stage. This is because during the raising stage, farmers have a high level of non-farm employment, which will reduce the time for farmers to collect traditional solid energy. In order to meet their daily energy needs, farmers will choose more convenient clean energy. Meanwhile, the family is in the scale expansion stage, and farmers are relatively young, confident in the future, with a high happiness index, and a strong ability to accept new things. This will result in a higher level of adoption of living clean energy. The family in the burdening stage has a significant negative impact on farmers' living clean energy adoption behavior, with disease being significant at a 5% level. It indicates that farmers during the burdening stage are unwilling to adopt living clean energy and their adoption level is significantly lower than that of the empty-nest stage. At this stage, the farmers' support burden is heavy, and they often choose not to use living clean energy for economic reasons. The family in the stable stage has a significant positive impact on farmers' living clean energy adoption behavior, with disease being significant at a 1% level. It indicates that farmers in the raising stage are willing to adopt living clean energy, and the degree of living clean energy adoption is significantly higher than that of empty-nest families. Farmers in this stage tend to engage in non-farm employment, with a relatively light economic burden and sufficient economic capital to adopt living clean energy. Farmers in the supported stage have a significant positive impact on farmers' living clean energy adoption behavior, with disease being significant at a 1% level. It indicates that farmers in the supported stage are willing to adopt living clean energy, and their adoption level is significantly higher than that of the empty-nest stage. This may be because farmers in this stage have higher health needs and a certain amount of capital accumulation, so they are more likely to adopt living clean energy.

From the perspective of the specific characteristic variables of the family life cycle, energy demand has a significant positive impact on farmers' living clean living energy adoption behavior at the 5% level. The higher the energy demand of farmers, the more traditional solid energy cannot meet their daily needs, which will encourage farmers to adopt living clean energy. However, this also affects farmers' adoption behavior when the original energy cannot meet the demand. Thus, energy demand only has a positive impact on farmers' living clean energy adoption behavior at the 5% level. Non-farm employment has a significant positive impact on farmers' living clean energy adoption behavior at the 1% level. The higher the level of non-farm employment of farmers, the less time they have to collect traditional energy. In order to meet the energy demand in their daily lives, it will increase the intensity of adopting living clean energy sources with convenient characteristics. Health demands have a significant positive impact on farmers' living clean energy adoption behavior at the 1% level. The higher the health demand, the more likely farmers are to adopt living clean energy that is less harmful to the human body. The ratio of expenditure to income has a significant negative impact on farmers' living clean energy adoption behavior at the level of 1%. The higher the ratio of expenditure to income, the heavier the support burden of the farmers, and they do not have a sufficient economic level to support their living clean energy adoption behavior. Therefore, the ratio of expenditure to income has a significant negative impact on farmers' living clean energy adoption behavior. In summary, the impact of the family life cycle on farmers' living clean energy adoption behavior is mainly reflected in four aspects: energy demand, livelihood strategy, health demand, and support burden. The higher the energy demand of farmers,

the higher the level of non-farm employment, the higher the health demand, and the lower the support burden, the higher the level of farmers' living clean energy adoption behavior.

In terms of controlling variables, first, regarding personal characteristics, gender has a significant negative impact on farmers' living clean energy adoption behavior, while age and education level have no significant impact. From the traditional perspective of gender differences, women often assume the role of family caregivers, so they are more likely to make more environmentally friendly consumption decisions due to the pursuit of maximizing family effects [61]. Secondly, in terms of cognitive factors, farmers' awareness of environmental ecology has a significant positive impact on their living clean energy adoption behavior, but their awareness of clean energy has no significant impact on their living clean energy adoption behavior. This may be because, at present, farmers may lack awareness of clean energy and not fully recognize the benefits of clean energy, resulting in an insignificant impact. Lastly, in terms of policy factors, the frequency of government promotion of living clean energy has a significant positive impact on farmers' living clean energy adoption behavior, while the satisfaction with relevant policies has no significant impact on farmers' behavior.

In reality, due to different purposes of use, there may be differences in the adoption behavior of farmers. Therefore, in order to investigate further, this paper, based on the measurement of farmers' living clean energy adoption behavior in the previous text, divides farmers' living clean energy adoption behavior into farmers' cooking, heating, and bathing clean energy adoption behavior according to different energy use purposes, and studies the impact of family life cycle on the three separately. The results are shown in Table 7, in which models 5, 6, and 7, respectively, take the farmers' cooking, heating and bathing clean energy adoption behavior as the dependent variable, respectively.

**Table 7.** Model results for the influence of family life cycle on farmers' cooking, heating, and bathing clean energy adoption behavior.

| Variable | Model 5 | | Model 6 | | Model 7 | |
|---|---|---|---|---|---|---|
| | B | S.E. | B | S.E. | B | S.E. |
| RS | 0.556 *** | 0.143 | 0.503 *** | 0.150 | 0.610 *** | 0.147 |
| BS | −0.356 * | 0.142 | −0.374 * | 0.149 | −0.278 | 0.146 |
| STS | 0.678 *** | 0.149 | 0.679 *** | 0.156 | 0.722 *** | 0.153 |
| SUS | 0.407 ** | 0.153 | 0.343 * | 0.160 | 0.518 *** | 0.157 |
| ED | 0.083 ** | 0.027 | 0.032 | 0.028 | 0.056 * | 0.028 |
| LS | 0.455 *** | 0.100 | 0.504 *** | 0.105 | 0.411 *** | 0.102 |
| HD | 0.068 * | 0.029 | 0.088 * | 0.031 | 0.067 * | 0.030 |
| SB | −0.113 * | 0.053 | −0.181 *** | 0.056 | −0.195 *** | 0.055 |
| X1 | −0.159 ** | 0.055 | −0.166 ** | 0.058 | −0.166 ** | 0.057 |
| X2 | 0.002 | 0.002 | 0.002 | 0.002 | 0.003 | 0.002 |
| X3 | 0.036 | 0.029 | 0.040 | 0.031 | 0.051 | 0.030 |
| X4 | 0.063 ** | 0.023 | 0.082 *** | 0.024 | 0.124 *** | 0.023 |
| X5 | 0.037 | 0.032 | 0.023 | 0.033 | 0.084 ** | 0.033 |
| X6 | 0.255 *** | 0.029 | 0.327 *** | 0.031 | 0.267 *** | 0.030 |
| X7 | −0.029 | 0.028 | −0.055 | 0.029 | −0.079 ** | 0.029 |
| Constant term | 0.786 ** | 0.277 | 0.747 ** | 0.291 | 0.559 | 0.285 |
| Adjusted R2 | 0.285 | | 0.286 | | 0.289 | |
| P | 0.000 | | 0.000 | | 0.000 | |

Note: B is the regression coefficient of the variable; *, **, and *** are statistically significant at the level of 1%, 5%, and 10%, respectively.

It can be seen from the table that in the family life cycle, except for the burdening stage, which has no significant impact on farmers' bathing clean energy adoption behavior, other family life cycle stages have a significant influence on farmers' cooking, heating, and bathing clean energy adoption behavior. In addition, in terms of farmers' cooking clean energy adoption behavior, energy demand, non-farm employment, health demand, awareness of the environment, and the frequency of government promotion all have a

positive impact on farmers' behavior and are significant at 5%, 1%, 10%, 5%, and 1% significance levels, respectively. Support burden and gender have a negative impact on farmers' behavior and are significant at 10% and 5% significance levels, respectively. In terms of farmers' heating clean energy adoption behavior, non-farm employment, health demand, awareness of the environment, and the frequency of government promotion all have a positive impact on farmers' behavior and are significant at 1%, 10%, 1%, and 1% significance levels, respectively. Support burden and gender have a negative impact on farmers' behavior and are significant at 1% and 5% significance levels, respectively. In terms of farmers' bathing clean energy adoption behavior, energy demand, non-farm employment, health demand, awareness of the environment, awareness of clean energy, and the frequency of government promotion all have a positive impact on farmers' behavior and are significant at 10%, 1%, 10%, 1%, 5%, and 1% significance levels, respectively. Support burden, gender, and satisfaction with policy have a negative impact on farmers' behavior and are significant at 1%, 5%, and 5% significance levels, respectively.

### 5.2. Analysis of Factors Influencing Farmers' Living Clean Energy Adoption Behavior at Different Family Life Cycle Stages

In order to further clarify the differences in influencing factors for farmers' living clean energy adoption behavior at different stages of their family life cycle, this paper conducts regression based on the data from farmers at different stages of the family life cycle. Due to the small number of samples in the raising stage and the empty-nest stage, the families in these two stages are not considered in this paper. The specific regression results are presented in Table 8. The *p*-values of the four-stage regression models were all 0.000, indicating that the model passed the 1% significance test. The adjusted R2 values were 0.278, 0.051, 0.200, and 0.239, respectively. This indicates that the overall goodness of fit of the model meets the requirements and is statistically significant.

**Table 8.** Influencing factors for farmers' living clean energy adoption behavior in different household life cycle stages.

| Variable | Raising Stage | | Burdening Stage | | Stable Stage | | Supported Stage | |
|---|---|---|---|---|---|---|---|---|
| | B | S.E. | B | S.E. | B | S.E. | B | S.E. |
| ED | 0.135 ** | 0.051 | 0.035 | 0.030 | 0.189 * | 0.080 | 0.070 | 0.082 |
| LS | 0.540 ** | 0.173 | 0.198 | 0.176 | 0.739 *** | 0.219 | 0.293 | 0.332 |
| HB | 0.169 *** | 0.050 | 0.071 | 0.037 | 0.105 | 0.083 | 0.030 | 0.087 |
| SB | −0.144 * | 0.068 | −0.272 *** | 0.075 | −0.052 | 0.233 | −0.401 | 0.316 |
| X1 | −0.249 ** | 0.094 | −0.011 | 0.068 | −0.446 ** | 0.154 | 0.049 | 0.184 |
| X2 | 0.006 | 0.005 | −0.002 | 0.003 | 0.009 | 0.008 | 0.000 | 0.007 |
| X3 | 0.090 | 0.052 | 0.005 | 0.035 | 0.063 | 0.089 | 0.059 | 0.094 |
| X4 | 0.141 *** | 0.040 | 0.015 | 0.026 | 0.120 | 0.071 | 0.248 ** | 0.079 |
| X5 | 0.095 | 0.060 | −0.070 | 0.037 | −0.021 | 0.091 | 0.064 | 0.098 |
| X6 | 0.413 *** | 0.053 | 0.124 *** | 0.033 | 0.373 *** | 0.094 | 0.470 *** | 0.088 |
| X7 | −0.099 * | 0.047 | −0.040 | 0.033 | −0.115 | 0.099 | −0.155 | 0.095 |
| Constant term | −0.131 | 0.429 | 1.947 *** | 0.303 | 0.687 | 0.786 | 0.406 | 0.767 |
| N | 448 | | 532 | | 202 | | 153 | |
| Adjusted R2 | 0.278 | | 0.051 | | 0.200 | | 0.239 | |
| P | 0.000 | | 0.000 | | 0.000 | | 0.000 | |

Note: B is the regression coefficient of the variable; *, **, and *** are statistically significant at the level of 1%, 5% and 10% respectively.

From the table, it can be seen that there is heterogeneity in the factors influencing farmers' living clean energy adoption behavior at different stages of the family life cycle.

During the raising stage, the main factors that significantly influence farmers' living clean energy behavior include energy demand, non-farm employment, health demand, gender, environmental awareness, and the frequency of government promotion. Among them, energy demand, non-farm employment, health demand, environmental awareness,

and the frequency of government promotion have a positive impact on farmers' adoption behavior at a significance level of 5%, 5%, 1%, 1%, and 1%, respectively. Support burden and gender have a negative impact on farmers' adoption behavior at a 10% and 5% significance level.

During the burdening stage, the main factors that significantly influence farmers' living clean energy behavior are the ratio of expenditure to income and the frequency of government promotion. The ratio of expenditure to income has a negative impact on farmers' adoption behavior at a significance level of 1%, while the frequency of government promotion has a positive impact on farmers' adoption behavior at a significance level of 1%. This is mainly because farmers at this stage have a heavy support burden and are more sensitive to economic costs, which often becomes the main reason for limiting their adoption of clean energy.

During the stable stage, the main factors that significantly influence farmers' living clean energy behavior include energy demand, non-farm employment, gender, and the frequency of government promotion. Energy demand, non-farm employment, and the frequency of government promotion all have a positive impact on farmers' adoption behavior at a significance level of 10%, 1%, and 1%, respectively. Gender has a negative impact on farmers' adoption behavior at a significance level of 5%, and government promotion frequency has a positive impact on farmers' adoption behavior at a significance level of 1%.

During the supported stage, the main factors that significantly influence farmers' living clean energy behavior include environmental awareness and the frequency of government promotion. Environmental awareness has a positive impact on farmers' adoption behavior at a significance level of 5%, while the frequency of government promotion has a positive impact on farmers' adoption behavior at a significance level of 1%.

We can see that, regardless of the stage, the frequency of government promotion of clean energy can positively affect farmers' living clean energy adoption behavior. This may be because living clean energy adoption requires a large amount of capital investment (purchase of boiler equipment, connection, construction of network infrastructure, etc.). So, the government subsidies and support are particularly important with respect to whether farmers adopt clean heating.

## 6. Conclusions

The promotion of living clean energy in rural areas is important for achieving China's carbon neutrality goal. China's rural areas are a "family-oriented" society, and farmers' decision making is often closely related to the whole family. At different stages of the family life cycle, the family has different family characteristics, which leads to heterogeneity in the focus and final decision of farmers to adopt living clean energy. Therefore, this paper analyzes farmers' living clean energy adoption behavior from the perspective of the family life cycle. This paper constructs a standard for classifying the family life cycle that is suitable for the actual situation in rural China. The rural family life cycle is divided into six stages: the initial stage, the raising stage, the burdening stage, the stable stage, the supported stage, and the empty-nest stage. On this basis, by using 1382 survey data collected in Jiangxi Province in 2022, this paper analyzed the influence mechanism of family life cycle on farmers' living clean energy adoption behavior and further explored the key influencing factors for farmers' living clean energy adoption behavior at different stages of the family life cycle. The study found the following:

(1) From the perspective of composite variables of the family life cycle, the raising stage, stable stage, and supported stage all have a significant positive impact on farmers' living clean energy adoption behavior, while the burdening stage has a significant negative impact on farmers' living clean energy adoption behavior;

(2) From the perspective of specific variables of the family life cycle, energy demand, non-farm employment, and health demand all have a significant positive impact

on farmers' living clean energy adoption behavior, whereas support burden has a significant negative impact on farmers' living daily clean energy adoption behavior;

(3)     Among the control variables, farmers' awareness of the surrounding environment and the frequency of government promotion of clean energy have a significant positive impact on their living clean energy adoption behavior. Gender has a significant negative impact on farmers' living clean energy adoption behavior;

(4)     At different stages of the family life cycle, there are both common and different factors that affect farmers' living clean energy adoption behavior. Regardless of the stage, the frequency of government promotion of clean energy can positively affect the farmers' living clean energy adoption behavior.

## 7. Discussion

Differently from most studies, this paper focuses on farmers' living clean energy adoption behavior in rural areas and uses the degree of adoption to comprehensively reflect their living clean energy adoption behavior. And based on the measurement of farmers' living clean energy adoption behavior in the previous text, this paper divides farmers' living clean energy adoption behavior into farmers' cooking, heating, and bathing clean energy adoption behavior according to different energy use purposes and studies the impact of family life cycle on the three separately. In addition, this paper not only uses composite variables of the family life cycle but also uses specific characteristic variables that can reflect the family life cycle according to specific representations in the process of farmers' living clean energy adoption behavior.

There is obvious heterogeneity of farmers' living clean energy adoption behavior in different family life cycles, indicating that farmers' living clean energy adoption decisions are closely related to the family life cycle. Therefore, the "one-size-fits-all" policy cannot meet the practical needs of farmers. When promoting clean energy in rural areas, the government should fully consider the different policy needs of farmers with different characteristics. For those at different stages of the family life cycle, policy formulation should be tailored to their specific resource endowments and development demands. This enables the government to provide targeted policy guidance and incentives to address the diverse needs of farmers at different stages of the family life cycle.

Meanwhile, it also should be noted that universal measures should be taken to address the common issues of farmers' living clean energy adoption behavior at different stages of their family life cycle. From the regression results of different family cycle stages, regardless of the stage, the frequency of government promotion of clean energy can positively affect farmers' living clean energy adoption behavior. Therefore, in the future promotion process, more attention should be paid to the role of government publicity and promotion. And it should be explained that the fundamental reason why government promotion has a positive impact on farmers' behavior is that government promotion can improve farmers' awareness and thus promote their behavior. In terms of cognitive factors, farmers' awareness of the environment can significantly and positively affect their adoption behavior, but their awareness of clean energy has no significant impact on their adoption behavior. This may be due to the fact that although farmers have gained a certain understanding of the environmental pollution currently facing rural areas under the strong promotion of the government, their existing understanding of clean energy is insufficient, and they cannot fully recognize the benefits of clean energy. Therefore, the impact is not significant.

In addition, the significant impact of non-farm employment and support burden on farmers' living clean energy adoption behavior is essentially the impact of farmers' economic conditions on their living clean energy adoption behavior. So, farmers can be encouraged to adopt living clean energy by providing them with economic support and reducing economic costs. To be specific, first of all, we can directly provide subsidies to farmers. Secondly, the price support system can be implemented, such as France's fixed electricity price subsidies for photovoltaic surplus electricity to reduce the cost of farmers' living clean energy adoption. Finally, the government can provide preferential policies

for clean energy companies. By injecting funds into related companies, it will reduce their production costs so as to further reduce the farmers' purchase cost of clean energy equipment. For example, in some Canadian provinces, local companies that invest in clean energy will be able to receive government subsidies ranging from CAD 20,000 to CAD 250,000.

The shortcomings of this paper are that, due to the limited sample size, no further in-depth research was conducted on farmers in the initial stage and the empty-nest stage. In addition, it only considered the farmers' living clean energy adoption behavior from the perspective of the degree of adoption. The long-term adoption behavior and the substitution effect between different energy sources may also affect farmers' adoption behavior, which were not considered in this paper. These will also be the direction and focus of the next research.

**Author Contributions:** Conceptualization, X.D.; methodology, X.D.; software, J.W.; validation, X.D. and S.L.; formal analysis, J.W.; investigation, X.D. and J.W.; resources, J.W.; data curation, X.D.; writing—original draft preparation, X.D.; writing—review and editing, J.W. and S.L.; visualization, X.D.; supervision, S.L.; project administration, S.L.; funding acquisition, S.L. All authors have read and agreed to the published version of the manuscript.

**Funding:** This research received no external funding.

**Institutional Review Board Statement:** Not applicable.

**Informed Consent Statement:** Informed consent was obtained from all subjects involved in the study.

**Data Availability Statement:** The data presented in this study are available on request from the corresponding author. The data are not publicly available due to data management.

**Conflicts of Interest:** The authors declare no conflict of interest.

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
