# Peer review of "The Impact of Family Life Cycle on Farmers’ Living Clean Energy Adoption Behavior—Based on 1382 Farmer Survey Data in Jiangxi Province"

_agriculture, doi:10.3390/agriculture13112084_

Round 1

Reviewer 1 Report

Comments and Suggestions for Authors

1) reference formating. 

2) Contribution is not very sharp in the abstract. Implication to be made clear and direct at point. Method to be detailed more in terms of variables to highlight novelty of the paper. 

3) Revise in terms of following concepts: rural developmen or agricultural development, food sustainability and food security in relation to sustainability.  

4)  Sectioning at the end of the introduction.

5) Literature review should be added after introduction, before method. 

6) Equations of regressions should be added to method.

7) Why Multiple regression, why not SEM? The model is well complex with links and what has been done with MLR method is similar to SEM with this respect. 

8) Descriptive statistics should be extended in Table 1 with skewness, kurtosis and relevant comments afterwards.

9) Is there references for data collection method, i.e., what is the scale source for ""How often you use elec- 299

tricity in the cooking process?", "How often you use solar energy in the cooking process?", 300

"How often you use natural gas in the cooking process?" and others? 

10) This is not correect and the value is too high: 

"VIF is less than 10, it is considered reasonable and there is no collinearity 396" 

Multicollinearity problems should be solved. 

11) DW test is for autocorrelation testing. Ideal value of it is in a range close to 2 depending on the sample size there are upper and lower DW limits. Interpretation of DW is wrong. 

12) Last section of the model is highly important but the results are mostly significant for Cognitive factors and Policy factors.

Generally at these situations, researchers also apply interaction between these factors. This could also help identify the effects of these factors. It should be tested. 

13) R2 should be revplaced with adjusted R2 for MLR models. (Multiple linear rgression)

14) Discussion with existent empirical literature with comparisons. 

15) limitations and future directions 

Comments on the Quality of English Language

Minor.

Reviewer 2 Report

Comments and Suggestions for Authors

However, this interesting study requires some correction.

Comments:

1. Please indicate what the purpose of the research is in the abstract. The goal indicated in the article is too general and incorrectly formulated (line 99-101)

2. What research gap has been identified and how does the article fill it?

3. The introduction about the family life cycle is too extensive. Some of the wording is too obvious (line 190-210).

4. How was the province divided into 3 levels? This is not clearly presented.

5. What does the following phrase mean: the sample households have a certain degree of representativeness (line 368-369)?

6. The factors indicated in the results require explanation. Why do individual variables affect the dependent variable in this way?

7. The conclusions are too detailed, they should be generalized and the importance and relationship of individual factors to the examined phenomenon should be indicated.

Reviewer 3 Report

Comments and Suggestions for Authors

Dear authors! The topic of introducing clean energy (especially in rural areas) is very important from both theoretical and practical points of view.

But your research raises more questions than it answers.

1. The title of the article talks about farmers. But in what follows we are talking only about households.

2. The authors only indicate in the middle of the article and only as an insignificant fact that we are talking about energy not related to the production activities of farmers.

3. The authors do not divide energy by purpose of use. Firewood, coal and straw can only be used for heating, cooking or bathing. The rest of the energy has no alternative. Greater widespread use of modern technology leads to an increase in the share of clean energy.

4. Energy consumption for heating, etc. depends primarily on family income, since the transition from a traditional heating system to a modern one requires significant capital investments (purchase of boiler equipment, connection, construction of network infrastructure, etc.). Without government assistance, this is a very expensive undertaking, especially in rural areas.

5. Non-agricultural activities of households, of course, greatly influence the possibility of using clean energy, since its profitability is always higher than agricultural ones.

6. The values of the approximation coefficients indicate that in all cases of application of the models the statistical relationship is very weak. This indicates that the provisions of the theory that the authors promote do not correspond to reality.

7. There are questions about the representativeness of the data. If we were talking about a general model, then the distribution of households into groups would be acceptable. But when they try to apply the models within individual groups, then 8 and 39 households are clearly not enough.

Round 2

Reviewer 1 Report

Comments and Suggestions for Authors

I had read the paper uploaded for the second round. I evaluated it in terms of two pillars, first is the previous critiques and how they were addressed, second is, the overall structure of paper, the sections, and the results derived, leading to discussion and policy recommendations. 

My comments for the second round are: 

1) Family Life Cycle (FLC) is central throughout the paper. The direction of focus is on FLC in 4 places in the abstract. As it is in the title, since it is the central independent variable as the relevant data section presented. 

Why is it so delayed to explain the importance and the discussion of the FLC in the introduction? Instead, main focus is given to policies, yes? However, FLC comes all the way down at the third page. This affects the structure of this section. My suggestion is to bring forth FLC. Also, the clean energy behaviour to the center with a similar respect. Discuss them in terms of their importance in agriculture, food security and sustainability impacts. 

2) Similarity should be reduced, preferably, lower than 15% or less, but most importantly, there are open sentences that Turnitin software refers to directly. Referencing at these places are important. 

3) Literature review is important and still it is not added. This section helps to identify where the location of the paper is. Without it, the novelty is also not clear to the readers. Literature review should be added. In discussion, findings should be evaluated in comparison with existent literature from different countries as well. 

4) Response 7 is not sound, by stating that SEM is not appropriate since the variables are continuous. Virtual variable is not an econometric nor statistical term to provide a rebuttal. SEM is a regression based model, that is similar in estimation techniques though it found application places in management related social sciences. The suggestion was just a suggestion, no need for rebuttal since the suggestion was not to add SEM results. The suggestion was not to add SEM results after a consideration such as that it was more adequate. More adequate methods exist for such data. The reasoning was to give insight for future readers.. 

3)  Discussion should include existent empirical papers and comparisons in the same / similar field of research. Not only in China, but rest of the world. How it was adapted in other countries? Which policies were applied? Which policies do authors suggest based on their findings for China? 

4) In conclusion, it was noted that at different levels of FLC, regardless of the stage, government promotion on clean energy is an important tool. However, it was noted in the previous sections (Intro.) that the responsibility should not be solely on the government but also on the farmers and all actors. The contrasting meanings should be cleared by discussing the results with these actors. Because, the following justification does not really address this farmer's involvement since these paragraphs still refer to the governement policies to improve say the education of the farmer. 

5) Introduction bases the focus on China's 2060 net zero target. Conclusion has not discussed this. For integrity, it is needed. 

6) In the previous round, limitations were asked and should be added. Authors responded with limitations. See below. But I cannot see a dedicated section in the conclusion with these respects. Also, no future directions are given. Though authors responded for this critique. These two issues need to be integrated to the paper. 

Point 15: limitations and future directions.

Response 15: Certainly, there are still some drawbacks and limitations to this study. First, due to the limited sample size, no further in-depth research was conducted on farmers in the initial stage and empty-nest stage. Secondly, it only considered the farmers’ living clean energy adoption behavior from the perspective of the degree of adoption. The long-term adoption behavior, and the substitution effect between different energy sources may also affect farmers’ adoption behavior, which was not considered in this paper. This will also be the direction and focus of the next research.

Comments on the Quality of English Language

Minor

Reviewer 3 Report

Comments and Suggestions for Authors

Dear authors! Thanks for a job well done.
